# Learning Latent Superstructures in Variational Autoencoders for Deep Multidimensional Clustering

**Xiaopeng Li[1], Zhourong Chen[1], Leonard K. M. Poon[2] and Nevin L. Zhang[1]**
[1] Department of Computer Science and Engineering
The Hong Kong University of Science and Technology
[2] Department of Mathematics & Information Technology
The Education University of Hong Kong
{xlibo,zchenbb,lzhang}@cse.ust.hk, kmpoon@eduhk.hk

## Abstract

We investigate a variant of variational autoencoders where there is a superstructure of discrete latent variables on top of the latent features. In general, our superstructure is a tree structure of multiple super latent variables and it is automatically learned from data. When there is only one latent variable in the superstructure, our model reduces to one that assumes the latent features to be generated from a Gaussian mixture model. We call our model the *latent tree variational autoencoder* (LTVAE). Whereas previous deep learning methods for clustering produce only one partition of data, LTVAE produces multiple partitions of data, each being given by one super latent variable. This is desirable because high dimensional data usually have many different natural facets and can be meaningfully partitioned in multiple ways.

## 1 Introduction

Clustering is a fundamental task in unsupervised machine learning, and it is central to many data-driven application domains. Cluster analysis partitions all the data into disjoint groups, and one can understand the structure of the data by examining examples in each group. Many clustering methods have been proposed in the literature (Aggarwal & Reddy, 2013), such as $k$-means (MacQueen et al., 1967), Gaussian mixture models (Christopher, 2016) and spectral clustering (Von Luxburg, 2007). Conventional clustering methods are generally applied directly on the original data space. However, it is challenging to perform cluster analysis on high dimensional and unstructured data (Steinbach et al., 2004), such as images. It is not only because the dimensionality is high, but also because the original data space is too complex to interpret, e.g. there are semantic gaps between pixel values and objects in images.

Recently, deep learning based clustering methods have been proposed that simultanously learn non-linear embeddings through deep neural networks and perform cluster analysis on the embedding space. The representation learning process learns effective high-level representations from high dimensional data and helps the cluster analysis. This is typically achieved by unsupervised deep learning methods, such as restricted Boltzmann machine (RBM) (Hinton et al., 2006; Hinton & Salakhutdinov, 2006), autoencoders (AE) (Vincent et al., 2008; 2010), variational autoencoders (VAE) (Kingma & Welling, 2014), etc. Previous deep learning based clustering methods (Xie et al., 2016; Guo et al., 2017; Jiang et al., 2017; Yang et al., 2017) assume one single partition over the data and that all attributes define that partition. In real-world applications, however, the assumptions are usually not true. High-dimensional data are often multifaceted and can be meaningfully partitioned in multiple ways based on subsets of attributes (Chen et al., 2012). For example, a student population can be clustered in one way based on course grades and in another way based on extracurricular activities. Movie reviews can be clustered based on both sentiment (positive or negative) and genre (comedy, action, war, etc.). It is challenging to discover the multi-facet structures of data, especially for high-dimensional data.

To resolve the above issues, we propose an unsupervised learning method, *latent tree variational autoencoder* (LTVAE) to learn latent superstructures in variational autoencoders, and simultaneously perform representation learning and structure learning. LTVAE is a generative model, where the data is assumed to be generated from latent features through neural networks, while the latent features themselves are generated from tree-structured Bayesian networks with another level of latent variables as shown in Fig. 1. Each of those latent variables defines a facet of clustering. The proposed method automatically selects subsets of latent features for each facet, and learns the dependency structure among different facets. This is achieved through systematic structure learning. Consequently, LTVAE is able to discover complex structures of data rather than one partition. We also propose efficient learning algorithms for LTVAE with gradient descent and Stepwise EM through message passing.

The rest of the paper is organized as follows. The related works are reviewed in Section 2. We introduce the proposed method and learning algorithms in Section 3. In Section 4, we present the empirical results. The conclusion is given in Section 5.

## 2 RELATED WORKS

Clustering has been extensively studied in the literature in many aspects (Aggarwal & Reddy, 2013). More complex clustering methods related to structure learning using Bayesian nonparametrics have been proposed, like Dirichlet Process (Blei et al., 2006), Hierarchical Dirichlet Process (HDP) (Teh et al., 2006). However, those are with conventional clustering methods that apply on raw data. Recently, deep learning based clustering methods have drawn more and more attention. A simple two-stage approach is to first learn low-dimensional embeddings using unsupervised feature learning methods, and then perform cluster analysis on the embeddings. However, without any supervision, the representation learning do not necessarily reveal the true cluster structure of the data. DEC (Xie et al., 2016) is a method that simultaneously learns feature representations and cluster assignments through deep autoencoders. It gradually improves the clustering by driving the deep network to learn a better mapping. Improved Deep Embedded Clustering (Guo et al., 2017) improves DEC by keeping the decoder network and adding reconstruction loss to the original clustering loss in DEC. Variational deep embedding (Jiang et al., 2017) is a generative method that models the data generative process using a Gaussian mixture model combined with a VAE, and also performs joint learning of representations and clustering. Similarly, GMVAE (Dilokthanakul et al., 2016) performs joint learning of a GMM and a VAE, but instead generates the mixture components through neural networks. Deep clustering network (DCN) (Yang et al., 2017) is another one that jointly learns an autoencoder and performs k-means clustering. These joint learning methods consistently achieve better clustering results than conventional ones. The method proposed in (Yang et al., 2016) uses convolutional neural networks and jointly learns the representations and clustering in a recurrent framework. All these methods assume flat partitions over the data, and do not attempt the structure learning issue. An exception is hierarchical nonparametric variational autoencoders proposed in (Goyal et al., 2017). It uses nCRP as the prior for VAE to allow infinitely deep and branching tree hierarchy structure and focuses on learning hierarchy of concepts. However, it is still one partition over the data, only that the partitions in upper levels are more general partitions, while those in lower levels more fine-grained. Different from it, our work focuses on multifacets of clustering, for example, the model could make one partition based on identity of subjects, while another partition based on pose.

## 3 THE PROPOSED METHOD

In this section, we present the proposed *latent tree variational autoencoder* and the learning algorithms for joint representation learning and structure learning for multidimensional clustering.

### 3.1 LATENT TREE VARIATIONAL AUTOENCODER

Deep generative models assume that data $\mathbf{x}$ is generated from latent continuous variable $\mathbf{z}$ through some random process. The process consists of two steps: (1) a value $\mathbf{z}$ is generated from some prior distribution $p(\mathbf{z})$; (2) the observation $x$ is generated from the conditional distribution $p_\theta(\mathbf{x}|\mathbf{z})$, which is parameterized through deep neural networks. Thus, it defines the joint distribution between

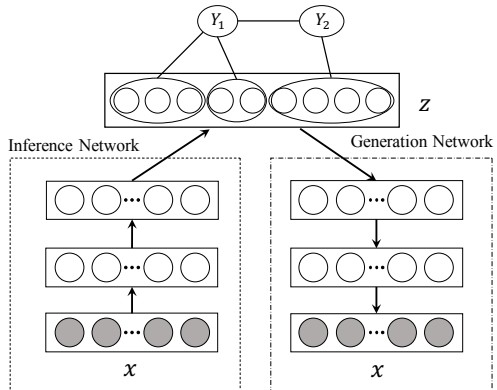

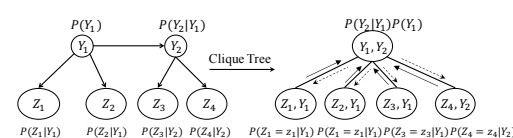

Figure 2: Inference and gradient through message passing. Solid-arrows denote collecting message, and dashed-arrows denote distributing message.

Figure 1: Latent Tree Variational Autoencoder

observation $\mathbf{x}$ and latent variable $\mathbf{z}$:

$$p(\mathbf{x}, \mathbf{z}) = p(\mathbf{z})p_\theta(\mathbf{x}|\mathbf{z}) \tag{1}$$

This process is hidden from our view, and we learn this process by maximizing the marginal log-likelihood $p(\mathbf{x})$ over the parameters $\theta$ and latent variable $\mathbf{z}$ from data. After the learning, the latent variable $\mathbf{z}$ can be regarded as the deep representations of $\mathbf{x}$ since it captures the most relevant information of $\mathbf{x}$. Thus, the learning process is also called representation learning.

In order to learn the latent structure of $\mathbf{z}$, for example multidimensional cluster structure, we introduce a set of latent variables $Y_1, ..., Y_l$ on top of $\mathbf{z}$. A single $z$ or multiple $z$'s form a node $\mathbf{z}_b$. Suppose variables in $\mathbf{z}$ form $B$ nodes of $\mathbf{z}_1, \cdots, \mathbf{z}_B$. Each latent variable $Y$ may be only connected to a subset of nodes, and the dependency of each $\mathbf{z}_b$ and its parent $Y$ is characterized by a conditional Gaussian distribution. Furthermore, the latent variables $Y_1, ..., Y_l$ are connected to each other, and the dependency of a latent variable $Y$ on its parent $Y'$ is characterized by a conditional distribution $P(Y|Y')$. This essentially forms a Bayesian network. And if we restrict the network to be tree-structured, the $\mathbf{z}$ and $\mathbf{Y}$ together form a *latent tree model* (Zhang, 2004; Poon et al., 2010; 2013; Mourad et al., 2013; Pearl, 2014; Zhang & Poon, 2017) with $\mathbf{z}$ being the observed variables and $\mathbf{Y}$ being the latent variables. For multidimensional clustering, each latent variable $Y$ is taken to be a discrete variable, where each discrete state $y$ of $Y$ defines a cluster. Each latent variable $Y$ thus defines a facet partition over the data based on subset of attributes and multiple $Y$'s define multiple facets. Given a value $y$ of $Y$, $\mathbf{z}_b$ follows a conditional Gaussian distribution $P(\mathbf{z}_b|y) = \mathcal{N}(\mu_y, \Sigma_y)$ with mean vector $\mu_y$ and covariance matrix $\Sigma_y$. Thus, each $\mathbf{z}_b$ and its parent constitute a Gaussian mixture model (GMM). Suppose the parent of a node is denoted as $\pi(\cdot)$, the maginal distribution of $\mathbf{z}$ is defined as follows

$$p(\mathbf{z}) = \sum_{\mathbf{Y}} \prod_{j=1}^{l} p(y_j|\pi(Y_j)) \prod_{b=1}^{B} \mathcal{N}(\mathbf{z}_b|\mu_{\pi(\mathbf{z}_b)}, \Sigma_{\pi(\mathbf{z}_b)}), \tag{2}$$

which sums over all possible combinations of $\mathbf{Y}$ states. As a matter of fact, a GMM is a Gaussian LTM that has only one latent variable connecting to all observed variables.

Let the latent structure of $\mathbf{Y}$ be $\mathcal{S}$, defining the number of latent variables in $\mathbf{Y}$, the number of discrete states in each variable $Y$ and the connectivity structure among all variables in $\mathbf{z}$ and $\mathbf{Y}$. And let the parameters for all conditional probabilities in the latent structure be $\Theta$. Both the latent structure $\mathcal{S}$ and the latent parameters $\Theta$ are unknown. We aim to jointly learn data representations and the latent structure. The proposed LTVAE model is shown in Fig. 1. The latent structure $\mathcal{S}$ are automatically learned from data and will be discussed in a later section.

Due to the existence of the generation network, the inference of the model is intratable. Instead, we do amortized variational inference for the latent variable $\mathbf{z}$ by introducing an inference network (Kingma & Welling, 2014) and define an approximate posterior $q_\phi(\mathbf{z}|\mathbf{x})$. The evidence lower bound (ELBO) $\mathcal{L}_{\text{ELBO}}$ of the marginal loglikelihood of the data given $(\mathcal{S}, \Theta)$ is:

$$\mathcal{L}_{\text{ELBO}}(\mathbf{x}) = \mathbb{E}_{q_\phi(\mathbf{z}|\mathbf{x})}[\log p_\theta(\mathbf{x}|\mathbf{z})] + \mathbb{E}_{q_\phi(\mathbf{z}|\mathbf{x})}[\log \sum_{\mathbf{y}} p_\mathcal{S}(\mathbf{z}, \mathbf{y}; \Theta)] + \mathbb{H}[q_\phi(\mathbf{z}|\mathbf{x})], \tag{3}$$

where $\log \sum_{\mathbf{y}} p_{\mathcal{S}}(\mathbf{z}, \mathbf{y}; \Theta)$ is the marginal loglikelihood of the latent variable $\mathbf{z}$ under the latent tree model, and $\mathbb{H}[\cdot]$ is the entropy. The conditional generative distribution $p_{\theta}(\mathbf{x}|\mathbf{z})$ could be a Gaussian distribution if the input data is real-valued, or a Bernoulli distribution if binary, parameterized by the generation network. Using Monte Carlo sampling, the ELBO can be asymptotically estimated by

$$\mathcal{L}_{\text{ELBO}}(\mathbf{x}) \simeq \frac{1}{M} \sum_{i=1}^{M} \log p_{\theta}(\mathbf{x}|\mathbf{z}^{(i)}) + \log \sum_{\mathbf{y}} p_{\mathcal{S}}(\mathbf{z}^{(i)}, \mathbf{y}; \Theta) + \mathbb{H}[q_{\phi}(\mathbf{z}|\mathbf{x})], \tag{4}$$

where $\mathbf{z}^{(i)} \sim q_{\phi}(\mathbf{z}|\mathbf{x})$. The term $\mathbb{H}[q_{\phi}(\mathbf{z}|\mathbf{x})]$ can be computed analytically if we choose the form of $q_{\phi}(\mathbf{z}|\mathbf{x})$ to be a Gaussian distribution $\mathcal{N}(\mathbf{z}; \mu_{\mathbf{x}}, \sigma_{\mathbf{x}})$: $\mathbb{H}[q_{\phi}(\mathbf{z}|\mathbf{x})] = \frac{J}{2} \log(2\pi) + \frac{1}{2} \sum_{j=1}^{J} (1 + \log \sigma_j^2)$, where $J$ is the dimensionality of $\mathbf{z}$.

Furthermore, the marginal loglikelihood $\log \sum_{\mathbf{y}} p_{\mathcal{S}}(\mathbf{z}^{(i)}, \mathbf{y}; \Theta)$ can be computed efficiently through message passing. Message passing is an efficient algorithm for inference in Bayesian networks (Koller & Friedman, 2009; Poon et al., 2013). In message passing, we first build a clique tree using the factors in the defined probability density. Because of the tree structure, each $\mathbf{z}_b$ along with its parent form a clique with the potential $\psi(\mathbf{z}_b, y)$ being the corresponding conditional distribution. This is illustrated in Fig. 2. With the sampled $\mathbf{z}^{(i)}$, we can compute the message $\psi'(y)$ by absorbing the evidence from $\mathbf{z}$. During collecting message phase, the message $\psi'(y)$ are sent towards the pivot. After receiving all messages, the pivot distributes back messages towards all $\mathbf{z}$. Both the posterior of $\mathbf{Y}$ and the marginal loglikelihood of $\mathbf{z}^{(i)}$ thus can be computed in the final normalization step.

## 3.2 PARAMETER LEARNING THROUGH GRADIENT DESCENT AND STEPWISE EM WITH MESSAGE PASSING

In this section, we propose efficient learning algorithms for LTVAE through gradient descent and stepwise EM with message passing.

Given the latent tree model $(\mathcal{S}, \Theta)$, the parameters of neural networks can be efficiently optimized through stochastic gradient descent (SGD). However, in order to learn the model, it is important to efficiently compute the gradient of the marginal loglikelihood $\log p_{\mathcal{S}}(\mathbf{z}; \Theta)$ from the latent tree model, the third term in Eq. 4. Here, we propose an efficient method to compute gradient through message passing. Let $\mathbf{z}_b$ be the variables that we want to compute gradient with respect to, and let $Y_b$ be the parent node. The marginal loglikelihood of full $\mathbf{z}$ can be written as

$$\log p_{\mathcal{S}}(\mathbf{z}; \Theta) = \log[\sum_{y_b} \mathcal{N}(\mathbf{z}_b | \mu_{y_b}, \Sigma_{y_b}) f(y_b)], \tag{5}$$

where $f(y_b)$ is the collection of all the rest of the terms not containing $\mathbf{z}_b$. The gradient $\mathbf{g}_{\mathbf{z}_b}$ of the marginal loglikelihood $\log p_{\mathcal{S}}(\mathbf{z}; \Theta)$ w.r.t $\mathbf{z}_b$ thus can be computed as

$$\mathbf{g}_{\mathbf{z}_b} = \frac{1}{p_{\mathcal{S}}(\mathbf{z}; \Theta)} \frac{\partial \sum_{y_b} f(y_b) \mathcal{N}(\mathbf{z}_b | \mu_{y_b}, \Sigma_{y_b})}{\partial \mathbf{z}_b} = \sum_{y_b} p(y_b | \mathbf{z}) \frac{\partial \log[f(y_b) \mathcal{N}(\mathbf{z}_b | \mu_{y_b}, \Sigma_{y_b})]}{\partial \mathbf{z}_b}$$

$$= \sum_{y_b} p(y_b | \mathbf{z}) \Sigma_{y_b}^{-1}(\mu_{y_b} - \mathbf{z}_b) \tag{6}$$

where $p(y_b | \mathbf{z})$ is the posterior probability of $y_b$ and can be computed efficiently with message passing as described in the previous section. The detailed derivation is in Appendix E. Since $\mathbf{z} = [\mathbf{z}_1, ..., \mathbf{z}_B]$, we have

$$\frac{\partial \log p(\mathbf{z})}{\partial \mathbf{z}} = \left[ \frac{\partial \log p(\mathbf{z})}{\partial \mathbf{z}_1}, ..., \frac{\partial \log p(\mathbf{z})}{\partial \mathbf{z}_B} \right] = [\mathbf{g}_{\mathbf{z}_1}, ..., \mathbf{g}_{\mathbf{z}_B}]. \tag{7}$$

With the efficient computation of the third term in Eq. 4 and its gradient w.r.t $\mathbf{z}$ through message passing, the parameters of inference network and generation network can be efficiently optimized through SGD.

In order to jointly learn the parameters of the latent tree $\Theta$, we propose Stepwise EM algorithm based on mini-batch of data. Specifically, we maximize the third term in Eq. 4, i.e. the marginal loglikelihood of $\mathbf{z}$ under the latent tree. In the Stepwise E-step, we compute the distributions $P(y, y'|\mathbf{z}, \theta^{(t-1)})$ and $P(y|\mathbf{z}, \theta^{(t-1)})$ for each latent node $Y$ and its parent $Y'$. In the Stepwise

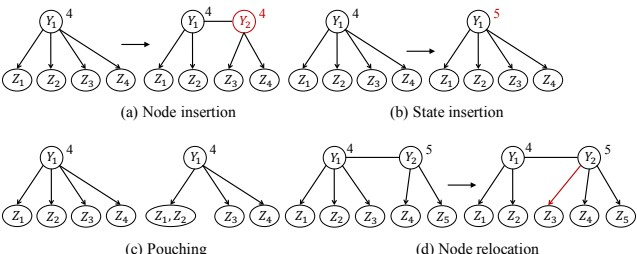

Figure 3: Structure search operators. The digits above the nodes denote the number of discrete states. Node deletion, state deletion and unpouching are the inverse of node insertion, state insertion and pouching, respectively.

M-step, we estimate the new parameter $\theta^{(t)}$. Let $\mathbf{s}(\mathbf{z}, \mathbf{y})$ be a vector the sufficient statistics for a single data case. Let $\bar{\mathbf{s}} = \mathbb{E}_{p_S(\mathbf{y}|\mathbf{z};\Theta)}[\mathbf{s}(\mathbf{z}, \mathbf{y})]$ be the expected sufficient statistics for the data case, where the expectation is w.r.t the posterior distribution of $\mathbf{y}$ with current parameter. And let $\mu = \sum_{i=1}^{N} \bar{\mathbf{s}}_i$ be the sum of the expected sufficient statistics. The update of the parameter $\Theta$ is performed as follows:

$$
\begin{aligned}
\bar{\mathbf{s}}_i^t &= \mathbb{E}_{p_S(\mathbf{y}_i|\mathbf{z}_i;\Theta^t)}[\mathbf{s}(\mathbf{z}_i, \mathbf{y}_i)] \\
\mu^{t+1} &= \mu^t + \eta(\bar{\mathbf{s}}_i^t - \mu^t) \\
\Theta^{t+1} &= \arg\max_{\Theta} l(\mu^{t+1}, \Theta),
\end{aligned}
\tag{8}
$$

where $\eta$ is the learning rate and $l$ is the complete data loglikelihood. Each iteration of update of LTVAE thus is composed of one iteration of gradient descent update for the neural network parameters and one iteration of Stepwise EM update for the latent tree model parameters with a mini-batch of data.

## 3.3 STRUCTURE LEARNING

For the latent structure $S$, there are four aspects need to determine: the number of latent variables, the cardinalities of latent variables, the connectivities among variables. We aim at finding the model $m^*$ that maximizes the BIC score (Schwarz et al., 1978; Koller & Friedman, 2009):

$$
BIC(m|\mathcal{D}) = \log P(\mathcal{D}|m, \theta^*) - \frac{d(m)}{2}\log N,
$$

where $\theta^*$ is the MLE of the parameters and $d(m)$ is the number of independent parameters. The first term is known as the likelihood term. It favors models that fit data well. The second term is known as the penalty term. It discourages complex models. Hence, the BIC score provides a trade-off between model fit and model complexity. To this end, we perform systematic searching to find a structure with a high BIC score. We use the hill-climing algorithm to search for $m^*$ as in (Poon et al., 2010; 2013), and define 7 search operators: node introduction (NI) and node deletion (ND) to introduce new latent nodes and delete existing nodes, state introduction (SI) and state deletion (SD) to add a new state and delete a state for existing nodes, node relocation (NR) to change links of existing nodes, pouching (PO) and unpouching (UP) operators to combine nodes into a single node and separate variables from a node.. The structure search operators are shown in Fig. 3. Each operator produces a set of candidates from existing structure, and the best candidate is picked if it improves the previous one. To reduce the number of possible search candidates, we first perform SI, NI and PO to expand the structure and pick the best model. Then we perform NR to adjust the best model. Finally, we perform UP, ND and SD to simplify the current best structure and pick the best one. Acceleration techniques (Poon et al., 2013) are adopted that make the algorithm efficient enough. The structure learning is performed iteratively together with the parameter learning of neural networks.

The overall learning algorithm is illustrated in Algorithm 1. Starting from a pretrained model, we iteratively improve the structure and parameters of latent tree model while learning the representations of data through neural network in a greedy manner. Using current structure $S^t$ as the initial structure,

---

**Algorithm 1** Learning Latent Tree Variational Autoencoder

---

**Input:** data $\mathcal{D}$, $\mathbf{z}$ dim, neural networks, $E$
$\theta, \phi, \mathcal{S}^0, \Theta^0 \leftarrow \text{pretrain}(\mathcal{D})$
**repeat**
    **for** $e = 1$ **to** $E$ **do**
        **for** each minibatch $\mathcal{X}$ in $\mathcal{D}$ **do**
            Compute $q_\phi(\mathbf{z}|\mu_\mathbf{x}, \sigma_\mathbf{x})$
            Sample $\mathbf{z}^{(i)} \sim q(\mathbf{z}|\mu_\mathbf{x}, \sigma_\mathbf{x})$
            Compute $\log p_\mathcal{S}(\mathbf{z}; \Theta)$ and $\frac{\partial \log p_\mathcal{S}(\mathbf{z})}{\partial \mathbf{z}}$ from Eq. 5 and 7
            Compute ELBO from Eq. 4
            $\theta, \phi \leftarrow$ Back-propagation and SGD step
            $\Theta \leftarrow \text{StepwiseEM}(\mathbf{z}^{(i)})$
        **end for**
    **end for**
    $\mathcal{D}_\mathbf{z} \leftarrow \mu_\mathcal{D}$
    **repeat**
        $\mathcal{S}^*, \Theta^* \leftarrow \text{SearchWith}(\mathcal{S}^{t-1}, \Theta^{t-1}, \{\text{SI, NI, PO}\})$
        $\mathcal{S}^*, \Theta^* \leftarrow \text{SearchWith}(\mathcal{S}^*, \Theta^* \{\text{NR}\})$
        $\mathcal{S}^t, \Theta^t \leftarrow \text{SearchWith}(\mathcal{S}^*, \Theta^*, \{\text{UP, ND, SD}\})$
    **until** $BIC(\mathcal{S}^t, \Theta^t | \mathcal{D}_\mathbf{z}) \leq BIC(\mathcal{S}^{t-1}, \Theta^{t-1} | \mathcal{D}_\mathbf{z})$
**until** stopping criteria
return $\theta, \phi, \mathcal{S}, \Theta$

---

we search for a better model. With new latent tree model, we optimize for a better representation until convergence.

## 4 EXPERIMENTS

### 4.1 SYNTHETIC-DATA DEMONSTRATION

We first demonstrate the effectiveness of the proposed method through synthetic data. Assume that the data points have two facets $Y_1$ and $Y_2$, where each facet controlls a subset of attributes (e.g. two-dimensional domain) and defines one partition over the data. This four-dimensional domain $\mathbf{z} = \{z_1, z_2, z_3, z_4\}$ is a latent representation which we do not observe. What we observe is $\mathbf{x} \in \mathbb{R}^{100}$ that is obtained via the following non-linear transformation:

$$\mathbf{x} = \sigma(U\sigma(W\mathbf{z})),$$

where $W \in \mathbb{R}^{10 \times 4}$ and $U \in \mathbb{R}^{100 \times 10}$ are matrices whose entries follow the zero-mean unit-variance i.i.d. Gaussian distribution, $\sigma(\cdot)$ is a sigmoid function to introduce nonlinearity. The generative model is shown in Fig. 4 (a). We define two clusters in facet $Y_1$ and two clusters in facet $Y_2$, and generate 5,000 samples of $\mathbf{x}$. Under the above generative model, recovering the two facets $Y_1$ and $Y_2$ structure and the latent $z$ domain from the observation of $\mathbf{x}$ seems very challenging. All previous DNN-based methods (AE+GMM, DEC, DCN, etc.) are only able to discover one-facet of clustering (i.e. one partition over the data), and none of these is applicable to solve such a multidimensional clustering problem. Fig. 4 (b) shows the results of the proposed method. As one can see, the LTVAE successfully discovers the true superstructure of $Y_1$ and $Y_2$. The 2-d plot of $z_1$ and $z_2$ shows the separable latent space clusters under facet $Y_1$, and it matches the ground-truth cluster assignments. Additionally, the 2-d plot of $z_3$ and $z_4$ shows another separable clusters under facet $Y_2$, and it also matches the ground-truth cluster assignments well in the other facet.

### 4.2 REAL-DATA EXPERIMENT SETUP

We evaluate the proposed LTVAE model on two image datasets and two other datasets, and compare it against other deep learning based clustering algorithms, including two-stage methods, AE+GMM and VAE+GMM, which first learn AE/VAE (Kingma & Welling, 2014) models then construct a GMM on top of them, and joint learning methods, DEC (Xie et al., 2016) and DCN (Yang et al., 2017). The datasets include MNIST, STL-10, Reuters (Xie et al., 2016; Jiang et al., 2017) and the Heterogeneity Human Activity Recognition (HHAR) dataset. When evaluating the clustering

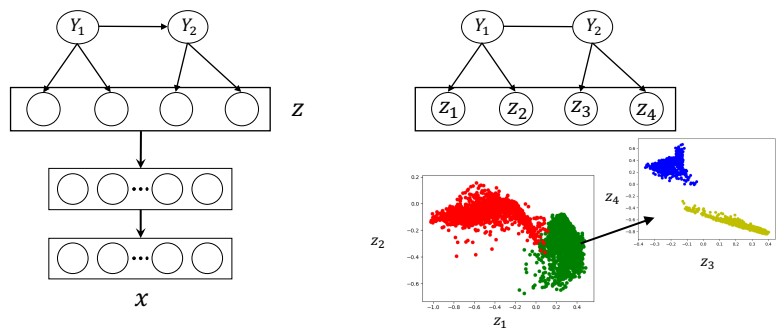

(a) Generative process for synthetic data

(b) Discovered superstructure and latent space

Figure 4: (a) The generative process of synthetic data; (b) The discovered multidimensional super-structure and the latent space (different colors denote different ground truth clusters in each facet.)

Table 1: Test data loglikelihood for various datasets.

| Model | MNIST | STL | Reuters | HHAR |
|-------|-------|-----|---------|------|
| VAE | -86.64±0.20 | -743.72±0.51 | -1312.40±1.24 | -17.88±0.26 |
| IWAE | -85.39±0.13 | -742.43±1.10 | -1254.29±6.95 | -16.53±0.16 |
| LTVAE | **-84.75±0.14** | **-619.04±7.88** | **-1245.71±4.45** | **-13.65±0.53** |

performance, for fair of comparison, we follow previous works (Xie et al., 2016; Yang et al., 2017) and use the network structures of $d-500-500-2000-10$ for the encoder network and $10-2000-500-500-d$ for the decoder network for all datasets, where $d$ is the data-space dimension, which varies among datasets. All layers are fully-connected. We follow the pretraining procedure as in (Xie et al., 2016). We first perform greedy layer-wise pretraining in denoising autoencoder manner, then stack all layers to form deep autoencoder. The deep autoencoder is further finetuned to minimize the reconstruction loss. The weights of the deep autoencoder are used to intialize the weights of encoder and decoder networks of above methods. After the pretraining, we optimze the objectives of those methods. For DEC and DCN, we use the same hyperparameter settings as the original papers. When initializing the cluster centroids for DEC and DCN, we perform 10 random restarts and pick the results with the best objective value for $k$-means/GMM. For the proposed LTVAE, we use Adam optimzer (Kingma & Ba, 2015) with initial learning rate of 0.001 and mini-batch size of 128. For Stepwise EM, we set the learning rate to be 0.01. As in Algorithm 1, we set $E = 5$, i.e. we update the latent tree model every 5 epochs. When optimizing the candidate models during structure search, we perform 10 random restarts and train with EM for 200 iterations.

## 4.3 TEST LOGLIKELIHOOD

We first show that, by using the marginal loglikelihood defined by the latent tree model as the prior, LTVAE better fits the data than conventional VAE and importance weighted autoencoders (IWAE) (Burda et al., 2016). While alternative quantitative criteria have been proposed (Bounliphone et al., 2016; Im et al., 2016; Salimans et al., 2016) for generative models, log-likelihood of held-out test data remains one of the most important measures of a generative model's performance (Kingma & Welling, 2014; Burda et al., 2016; Wu et al., 2017; Goyal et al., 2017). For comparison, we approximate true loglikelihood $\mathcal{L}_{5000}$ using importance sampling (Burda et al., 2016): $\mathcal{L}_k(\mathbf{x}) = \log \frac{1}{k} \sum_{i=1}^{k} \frac{p_\theta(\mathbf{x}, \mathbf{z}^{(i)})}{q_\phi(\mathbf{z}^{(i)}|\mathbf{x})}$, where $\mathbf{z}^{(i)} \sim q_\phi(\mathbf{z}|\mathbf{x})$. The results for all datasets are shown in Table 1. The proposed LTVAE obtains a higher test data loglikelihood and ELBO, implying that it can better model the underlying complex data distribution embedded in the image data.

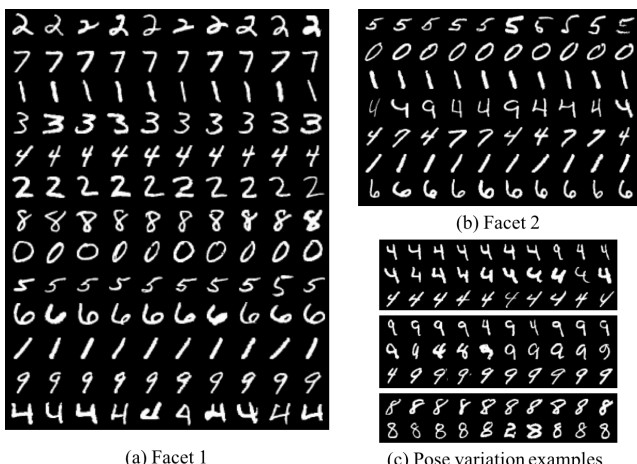

(a) Facet 1     (b) Facet 2     (c) Pose variation examples

Figure 5: Two facet clustering results from LTVAE are in (a) digit identity and (b) shape and pose. Each row contains the top 10 scoring elements from one cluster. (c) shows the pose variations by fixing the digit cluster in facet 1 and changing the cluster in facet 2. It can be seen that up-right, left-tilted and right-tilted images of the same digits are clearly recognizable.

Table 2: Clustering Accuracy of clustering results.

| Model | MNIST | STL-10 | Reuters | HHAR |
|---|---|---|---|---|
| AE+GMM | 82.18% | 79.83% | 68.68% | 78.90% |
| VAE+GMM | 76.87% | 79.49% | 65.85% | 67.91% |
| DEC | 84.30% | 80.62% | 74.32% | 79.86% |
| DCN | 83.32% | 85.88% | 75.05% | 81.26% |
| LTVAE | **86.32%** | **90.00%** | **80.96%** | **85.00%** |

## 4.4 MULTIFACET CLUSTERING

The most important features of the proposed model are that it can perform variable selection for model-based clustering, leading to multiple facets clustering.

We use the standard unsupervised evaluation metric and protocols for evaluations and comparisons to other algorithms (Yang et al., 2010). For baseline algorithms we set the number of clusters to the number of ground-truth categories. While for LTVAE, it automatically determines the number of facets and latent superstructure through structure learning. We evaluate performance with *unsupervised clustering accuracy (ACC)*:

$$ACC = \max_m \frac{\sum_{i=1}^n \mathbf{1}\{l_i = m(c_i)\}}{n},$$

where $l_i$ is the groundtruth label, $c_i$ is the cluster assignment produced by the algorithm, and $m$ ranges over all possible mappings between clusters and labels. Table 2 show the quantitative clustering results compared with previous works. With **z** dimension of small value like 10, LTVAE usually discovers only one facet. It can be seen the, for MNIST dataset LTVAE achieves clustering accuracy of 86.32%, better than the results of other methods. This is also the case for STL-10, Reuters and HHAR.

More importantly, the proposed LTVAE does not just give one partition over the data. Instead, it explains the data in multi-faceted ways. Unlike previous clustering experiments, for this experiment, we choose the **z** dimension to be 20. Fig. 5 shows the two facet clustering results for MNIST. It can be seen that facet 1 gives quite clean clustering over the identity of the digits and the ten digits are well separated. On the other hand, facet 2 gives a more grand partition based on the shape and pose. Note how up-right "4" and "9" are similar, and how tilted "4","7" and "9" are similar. The facet meanings are more evident in Fig. 5 (c). Fig. 6 shows four facets discovered for the STL-10

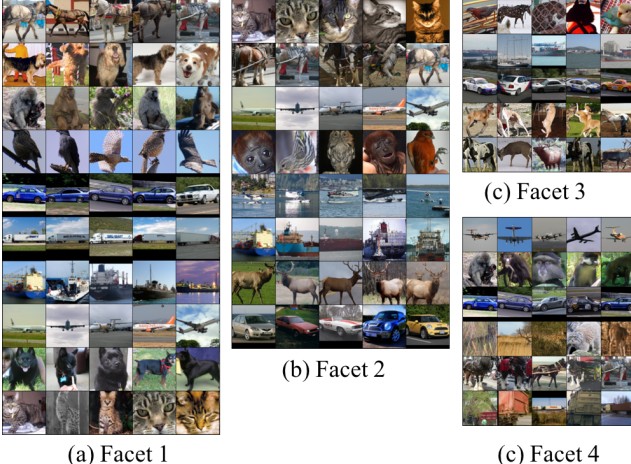

(c) Facet 3

(b) Facet 2

(a) Facet 1

(c) Facet 4

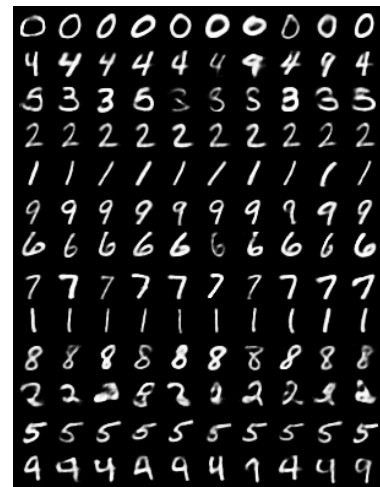

Figure 6: Clustering results from LTVAE for STL-10 dataset. Each row contains the top 5 scoring elements from one cluster.

Figure 7: The digits generated by the proposed model. Digits in the same row come from the same latent code of the latent tree.

dataset. Although it is hard to characterize precisely how the facets differ from each other, there are visible patterns. For example, the cats, monkeys and birds in facet 2 have clearly visible eyes, while this is not always true in facet 1. The deers in facet 2 are all showing their antlers/ears, while this is not true in facet 3. In facet 2 we see frontal views of cars, while in facets 1 and 3 we see side view of cars. In facet 1, each cluster consists of the same types of objects/animal. In facet 3/4, images in the same cluster do not necessarily show the same type of objects/animals. However, they have similar overall feel.

### 4.5 IMAGE GENERATION

Since the structure of the data in latent space is automatically learned through the latent tree, we can sample the data in a more structured way. One way is through ancestral sampling, where we first sample the root of the latent tree and then hierarchically sample the children variables to get $\mathbf{z}$, from which the images can be generated through generation network. The other way is to pick one component from the Gaussian mixture and sample $\mathbf{z}$ from that component. This produces samples from a particular cluster. Fig. 7 shows the samples generated in this way. As it can be seen, digits sampled from each component has clear semantic meaning and belong to the same category. Whereas, the samples generated by VAE does not have such structure. Conditional image generation can also be performed to alter the attributes of the same digit as shown in Appendix B.

## 5 DISCUSSIONS

LTVAE learns the dependencies among latent variables $\mathbf{Y}$. In general, latent variables are often correlated. For example, the social skills and academic skills of a student are generally correlated. Therefore, its better to model this relationship to better fit the data. Experiments show that removing such dependencies in LTVAE models results in inferior data loglikelihood.

In this paper, for the inference network, we simply use mean-field inference network with same structure as the generative network (Kingma & Welling, 2014). However, the limited expressiveness of the mean-field inference network could restrict the learning in the generative network and the quality of the learned model (Webb et al., 2018; Rainforth et al., 2018; Cremer et al., 2018). Using a faithful inference network structure as in (Webb et al., 2018) to incorporate the dependencies among latent variables in the posterior, for example one parameterized with masked autoencoder distribution estimator (MADE) model (Germain et al., 2015), could have a significant improvement in learning. We leave it for future investigation.

## 6 CONCLUSIONS

In this paper, we propose an unsupervised learning method, latent tree variational autoencoder (LT-VAE), which simultaneously performs representation learning and multidimensional clustering. Different from previous deep learning based clustering methods, LTVAE learns latent embeddings from data and discovers multi-facet clustering structure based on subsets of latent features rather than one partition over data. Experiments show that the proposed method achieves state-of-the-art clustering performance and reals reasonable multifacet structures of the data.

### ACKNOWLEDGMENTS

Research on this article was supported by Hong Kong Research Grants Council under grants 16212516 and 16202118.

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

## A SUPERSTRUCTURES

For the MNIST dataset, the conditional probability between identity facet $Y_1$ (x-axis) and pose facet $Y_2$ (y-axis) is shown in Fig. 8. It can be seen that a cluster in $Y_1$ facet could correspond to multiple clusters in $Y_2$ facet due to the conditional probability, e.g. cluster 0, 4, 5, 11 and 12. However, not all clusters in $Y_2$ facet are possible for a given cluster in $Y_1$ facet.

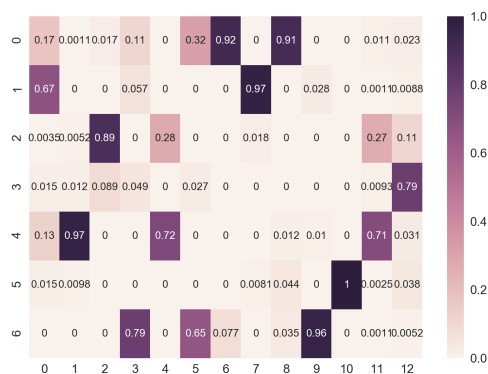

Figure 8: Conditional probability of $Y_1$ and $Y_2$ for the two facets of MNIST discovered by LTVAE.

## B CONDITIONAL IMAGE GENERATION

Here we show more results on conditional image generation. Interestingly, with LTVAE, we can change the original images by fixing variables in some facet and sampling in other facets. For example, in MNIST we can fix the variables in identity facet and change the pose of the digit by sampling in the pose facet. Fig. 9 shows the samples generated in this way. As it can be seen, the pose of the input digits are changed in the samples generated by the proposed method.

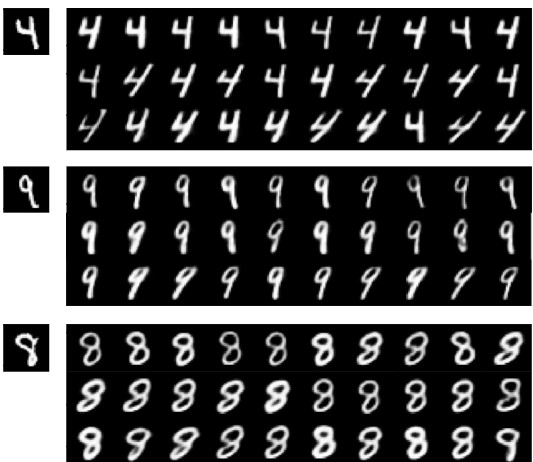

Figure 9: Image generation. Left are the original image. Right are generated with the proposed model by fixing the variables in identity facet and sampling the variables in the pose facet. Digits in the same row come from the same latent code of the latent tree.

## C   COMPUTATIONAL TIME

We compare the computational time of the proposed LTVAE w/ structure learning and that w/ fixed structure. For LTVAE with fixed structure, we fixed the structure of the latent tree model to be a single $Y$ connecting to all $z$s, in which each $z$ node consists of single $z$ variable.

Table 3: Computational time (s) w/ and w/o structure learning.

| Model | MNIST | STL-10 | Reuters | HHAR |
|---|---|---|---|---|
| LTVAE w/ structure learning | 7,592 | 3,251 | 5,756 | 5,693 |
| LTVAE w/ fixed structure | 3,197 | 542 | 1,442 | 1,021 |

## D   SIMILARITY BETWEEN LEARNED FACETS

Different facets learned by LTVAE might have some overlaps among each other. Here we make quantitative comparison among different facets based on the cluster assignments in each facet. We evaluate the similarity between two clusterings $Y_1$ and $Y_2$ using normalized mutual information $NMI(Y_1; Y_2)$. The NMI is given by

$$NMI(Y_1; Y_2) = \frac{I(Y_1; Y_2)}{\sqrt{H(Y_1)H(Y_2)}},$$

where $I(Y_1; Y_2)$ is the mutual information between $Y_1$ and $Y_2$ and $H(V)$ is the entropy of a variable $V$. These quantities can be computed from $P(Y_1; Y_2)$, which in turn is estimated by $P(Y_1; Y_2) = \frac{1}{N} \sum_{i=1}^{N} P(Y_1|\mathbf{d_i})P(Y_2|\mathbf{d_i})$, where $\mathbf{d}_1, \cdots, \mathbf{d}_N$ are the samples in the test data.

Table 4: NMI between different facets learned by LTVAE and groundtruth for MNIST dataset.

| | Groundtruth | Facet 1 | Facet 2 |
|---|---|---|---|
| Groundtruth | 1 | 0.825 | 0.574 |
| Facet 1 | 0.825 | 1 | 0.682 |
| Facet 2 | 0.574 | 0.682 | 1 |

Table 5: NMI between different facets learned by LTVAE and groundtruth for STL dataset.

| | Groundtruth | Facet 1 | Facet 2 | Facet 3 | Facet 4 |
|---|---|---|---|---|---|
| Groundtruth | 1 | 0.8613 | 0.6279 | 0.5758 | 0.5536 |
| Facet 1 | 0.8613 | 1 | 0.6962 | 0.6314 | 0.6251 |
| Facet 2 | 0.6279 | 0.6962 | 1 | 0.4675 | 0.5031 |
| Facet 3 | 0.5758 | 0.6314 | 0.4675 | 1 | 0.5885 |
| Facet 4 | 0.5536 | 0.6251 | 0.5031 | 0.5885 | 1 |

# E    DERIVATION OF GRADIENT

Here we give detailed derivation of Equation 6. The gradient $\mathbf{g}_{\mathbf{z}_b}$ of the marginal loglikelihood $\log p_{\mathcal{S}}(\mathbf{z}; \Theta)$ w.r.t $\mathbf{z}_b$ thus can be computed as

$$
\begin{aligned}
\mathbf{g}_{\mathbf{z}_b} &= \frac{\partial \log p_{\mathcal{S}}(\mathbf{z}; \Theta)}{\partial z_b} \\
&= \frac{1}{p_{\mathcal{S}}(\mathbf{z}; \Theta)} \frac{\partial p_{\mathcal{S}}(\mathbf{z}; \Theta)}{\partial z_b} \\
&= \frac{1}{p_{\mathcal{S}}(\mathbf{z}; \Theta)} \frac{\partial \sum_{y_b} f(y_b) \mathcal{N}(\mathbf{z}_b | \mu_{y_b}, \Sigma_{y_b})}{\partial \mathbf{z}_b} \\
&= \sum_{y_b} \frac{1}{p_{\mathcal{S}}(\mathbf{z}; \Theta)} \frac{\partial [f(y_b) \mathcal{N}(\mathbf{z}_b | \mu_{y_b}, \Sigma_{y_b})]}{\partial \mathbf{z}_b} \qquad (9)\\
&= \sum_{y_b} \frac{f(y_b) \mathcal{N}(\mathbf{z}_b | \mu_{y_b}, \Sigma_{y_b})}{p_{\mathcal{S}}(\mathbf{z}; \Theta)} \frac{\partial \log [f(y_b) \mathcal{N}(\mathbf{z}_b | \mu_{y_b}, \Sigma_{y_b})]}{\partial \mathbf{z}_b} \\
&= \sum_{y_b} p(y_b | \mathbf{z}) \frac{\partial \log [f(y_b) \mathcal{N}(\mathbf{z}_b | \mu_{y_b}, \Sigma_{y_b})]}{\partial \mathbf{z}_b} \\
&= \sum_{y_b} p(y_b | \mathbf{z}) \Sigma_{y_b}^{-1} (\mu_{y_b} - \mathbf{z}_b)
\end{aligned}
$$

where $p(y_b | \mathbf{z})$ is the posterior probability of $y_b$ and can be computed efficiently with message passing as described in the previous section. Note that

$$
p(y_b | \mathbf{z}) = \frac{f(y_b) \mathcal{N}(\mathbf{z}_b | \mu_{y_b}, \Sigma_{y_b})}{p_{\mathcal{S}}(\mathbf{z}; \Theta)} \qquad (10)
$$

is valid due to $p_{\mathcal{S}}(\mathbf{z}; \Theta) = \sum_{y_b} \mathcal{N}(\mathbf{z}_b | \mu_{y_b}, \Sigma_{y_b}) f(y_b)$ and the Bayes rule.

