# OpenReview forum: "Learning Latent Superstructures in Variational Autoencoders for Deep Multidimensional Clustering"
_ICLR.cc/2019/Conference_

### Official Review · AnonReviewer1 · 2018-10-16

**Rating:** 7
**Confidence:** 3

**Review:**

Revision post-discussion: The paper's notation and model has been clarified, and my concerns about the paper have been addressed. Proposing a latent tree structure on the latent space of generative models is a strong contribution, the model performs well and seems to find meaningful and interpretable structure in the latent space.


The paper proposes a latent tree superstructure for the latent space of VAE’s. The idea itself is novel and interesting, and could have major impact in learning structured manifolds.

The overall presentation of the method is direct but slightly confusing. It seems that the zb grouping corresponds to different dimensions of the full z_i-vector of a single data point x_i. This should be made more explicit.

The method itself has three levels of groupings: the zb’s, the conditioned variables Yb, and the connections between the Y’s. The method is also called a  Bayesian Network, but the paper seems to avoid defining it as a BN. I wonder if the method could be presented in a simpler form, if all the structure is necessary, and if the method could be defined directly as a BN. For instance, why do the Y’s have to have a hierarchical tree structure, wouldn’t a “flat” grouping into zb's be sufficient?

In eq 2 the p(z) is defined as a mixture of Y-conditioned Gaussians, while in eq 4 its defined in the conventional encoder form N(z ; mu_x, sigma_x). These forms don’t seem to be compatible with each other. The term H seems to be entropy, but its not explained. It can’t be computed if we use the eq 2 definition of p(z). The interplay between these two structures is unclear. Furthermore, in fig 1 the tree is showed as a network (no arrows), while in fig 2 its a tree. I can’t find the definition for the dependencies P(Y | Y’), are these simply conditional density tables, or are they implicit? I also can’t see how are the \Sigma_{yb} defined. Are they of full rank? What is their dimension?

The inference sections are well motivated and efficient techniques are used.

The synthetic experiment has 4 dimensional “z”, but the “W” matrix is 10x2, these do not match. What is the connection between Y_1 and Y_2 (in fig4 there is a dependency between)? Why is the dependency undirected if the model is a tree? The fig4b does not show ground truth to assess how well the model fits. The experiment should also include comparisons to the mentioned earlier works, and show how they perform. Why is there an arrow from the green scatter to the z3/z4? The main problem of the synthetic example is that it does not demonstrate why the tree structure learning is useful. The experiment should highlight a case where there is a natural latent tree structure corresponding to some realistic phenomena in real datasets.

The section 4.3. shows that the proposed method does find better representations of the MNIST than VAE, but does not mention that there are numerous extended VAE methods (and others) that would perform better than the LTVAE here. Those should be at least acknowledged, and preferably compared to.

The main results of the paper are very good with great performance in clustering, and the facets and clusters look great. The system has clearly learnt meaningful latent structures.

There are no learning curves or running time analyses. One would expect the proposed method to be slow with multiple levels of inference (tree structure, tree parameters, AE networks), and this should be discussed. How large datasets can it handle?

Overall the paper proposes a BN-style structure on VAE latent space with great performance, but somewhat incomplete experimental section, and some presentation issues.

---

> ### Author Response · Authors · 2018-11-23
> **Authors' response to reviewer 1**
>
> Q: The paper seems to avoid defining it as a BN. Could it be presented in a simpler form? Why do the Y’s have to have a hierarchical tree structure, wouldn’t a “flat” grouping into z_b’s be sufficient?
> A: The latent tree part of the model is, as stated in the Introduction of page 2, a tree-structured Bayesian network, where the internal variables are all latent. The paper doesn’t avoid defining it as a BN, but rather use “latent tree” to emphasize such property.
> In our model, Y’s need to have a hierarchical tree structure. One latent variable corresponds to one facet of data. In general, latent variables are often correlated. For example, the social skills and academic skills of a student are often correlated. Therefore, it’s better to model this relationship to better fit the data.
> We further show that models with no links between Y_1 & Y_2 are inferior to models with links. We take the model structure produced by LTVAE, remove the links between Y1 & Y_2 and it becomes two independent Gaussian mixtures. We re-estimate the parameters of the resulting model, and evaluate the test-set log-likelihood of it. The test-set log-likelihood of such model in MNIST is -120, which is significantly lower than that of LTVAE. It shows that the independent Gaussian mixtures model does not fits the data as good as LTVAE, suggesting the importance of hierarchical structure on the Y variables
>
> Q: The form of p(z) and the computation of entropy H.
> A: During modeling, we have p(z) as the prior distribution of z and p(x|z) as the generative distribution. The model is intractable. Therefore, during inference, we must introduce a variational distribution q(z|x) in order to do variational inference to approximate the true posterior. Note that it is q(z|x) that has the form of N(z; mu_x, sigma_x), and we do Monte Carlo sampling to make the inference. Yes, H is the entropy, and we now added its explanations. As in Eq.4, the entropy term is w.r.t q(z|x), instead of p(z). Therefore, it is computable.
>
> Q: The tree structure in Fig. 1 and Fig. 2 with arrows or not with arrows? The definition for the dependencs P(Y|Y’) and \Sigma_{y_b}.
> A: It should be noted that the edge orientations in LTMs are unidentifiable, meaning that the model with Y_1 being the parent of Y_2 is equivalent with the model with Y_2 being the parent of Y_1. It makes more sense to talk about undirected LTM, however, in implementation, an undirected model is represented using an arbitrary directed model in the equivalence class it represents.
> Therefore, in Fig. 1, we show the network without arrows between Ys. While in Fig. 2, we are showing how the message passing is conducted, thus deals with a directed implementation of the network.
> Since Ys are discrete variables, the conditional probability P(Y|Y’) is simply a conditional probability table. They are explicitly learned through StepwiseEM as in Section 3.2.
> A node z_b could be formed by a single or multiple z’s. The parent Y of node z_b form a Gaussian mixture, of which each component is a Gaussian N(\mu_yb, \Sigma_yb). \Sigma_yb is full-rank covariance matrix. The dimension of it depends on how many z’s are in the node z_b, whose structure is learned by structure learning.
>
> Q: The synthetic experiment, “W” matrix is 10x2. In synthetic experiment, what is the connection between Y_1 and Y_2 and why it is undirected if the model is a tree?
> A: We are sorry about the typo. W matrix is 10x4. We have stated the unidentifiability of the edge orientations in LTMs above. In the synthetic experiment, indeed, the connection between Y_1 and Y_2 should be directed, and we have fixed it in the updated pdf.
>
> Q: Groundtruth in Fig. 4b.
> A: In Fig. 4b, different color denotes the groundtruth label. It can be seen that in z_1, z_2 space, the two clusters are well separated. Looking further into the green cluster, the arrow zooms in the green points on z_3, z_4 space. The colors in z_3, z_4 space also show the groundtruth label in that facet.
>
> Q: Section 4.3, compare with other methods for the loglikelihood.
> A: We add the loglikelihood results of IWAE (Burda et al., 2016) in Table 6 in Appendix E for additional baseline comparison. Note that IWAE is more like a training method with loss function closer to loglikelihood than that of VAE. LTVAE could potentially also be improved with such training method. Other baseline methods in this paper cannot compute data loglikelihood.
>
> Q: There are no learning curves or running time analysis. How large datasets can it handle?
> A: The learning of the model mainly based on stochastic updates with mini-batch data. In principle, it is able to scale up to large datasets. We provide the computational time analysis in Appendix C in the updated pdf. We compare the computational time of LTVAE with structure learning and that with fixed structure. The structure learning costs several times more time than the fixed structure. The learning could be sped up significantly with GPU implementation.

---

> > ### Comment · AnonReviewer1 · 2018-11-26
> > **concerns addressed**
> >
> > Thanks for the explanations. My concerns about the paper have been resolved based also on the other responses, and I think this is a great paper that should be published at ICLR. I’m improving my rating to 7.
> >
> > It seems the main benefit of the approach is introduced structured dependencies on the latent space, where different poses of the digits are grouped under the conditioned identity facet, instead of being some arbitrarily shaped region in the standard MNIST-VAE latent space. It would be interesting to have discussion about the wider applicability of this.
> >
> > I share the concern with R3 about the non-clustered MNIST log-likelihood. -120 is bad value, and indicates issues with optimisation. This argument is not convincing. To me this paper is mainly motivated by improved interpretability instead of improved model fits.

---

### Official Review · AnonReviewer3 · 2018-10-30
**Interesting approach, but experiments could have a more in-depth analysis**

**Rating:** 7
**Confidence:** 4

**Review:**

The authors propose to augment the Variational AutoEncoder [1] with a latent prior modeled by a Gaussian Latent Tree Model [2], allowing to introduce a hierarchical structure of clusters in the learned representation. The LT-VAE not only learns the location of each cluster to best represent the data, but also their number and the hierarchical structure of the underlying tree. This is achieved by a three-step learning algorithm. Step 1 is a traditional training of the encoder and decoder neural networks to improve their fitting of the data. Step 2 is an EM-like optimization to better fit the parameters of latent prior to the learned posterior. And step 3 adapts the structure of the latent prior to improve its BIC score [3], which balances a good fit of the latent posterior with the number of parameter (and thus complexity) of the latent prior.

Experiments on synthetic data confirms the ability of the model to discover latent multifaceted clustering, and tests on 4 datasets shows it to be competitive with other unsupervised clustering models. Qualitative interpretation of samples from the learned model shows that the model learns a clustering that is clearly relevant to the data, while maybe not obvious to interpret.

The paper is well written and easy to follow (I however found a few typos and small mistakes that I'll list at the end of this review). The idea of using a structure on the latent prior of a VAE to learn a clustering of the data is not new, but the authors propose here an interesting approach to it, with a clearly described algorithm.

However, I would have liked to see a more in-depth analysis of the behavior of the model on the various datasets, and my reading of this paper raised several questions that found no answer:

1. What gains does the hierarchical structure on the Y variables provide? The paper does not analyze whether the models they trained actually learned conditional dependencies on the Y_i variables. How would this compare to the same model, with the only difference that the Y_i are fixed to be independent of each other (but still learning the number of Y_i and how the z_j are distributed between them) ?

2. This is linked to the previous one. On the tests of the dataset, how do the different facets interact with each other? How are the samples from the different clusters of facet 2 when facet 1 is fixed to a particular cluster? Assuming the learned dependency is that Y_1 is the parent of Y_2, does the interpretation of each value of Y_2 change depending on the value of Y_1?

3. The VAE with diagonal gaussian latent has a natural tendency to achieve sparcity in its latent space [4], making it robust to having too many latent neurons. Does this property hold with LT-VAE? If so, are the "unused" neurons organized in a particular way among the different learned facets?

I'd be reluctant to accept this paper without answers to points 1 and 2, which in my opinion are needed to justify the "tree" part of the "latent tree model" choice for the latent space. I'd also be very interested in an answer to point 3, which would give good insights regarding the design choices for applying this model to new problems (how important is the choice of the size of the latent space?), but I'm not considering it blocking acceptance.

[1] https://arxiv.org/abs/1312.6114
[2] http://jmlr.org/papers/volume5/zhang04a/zhang04a.pdf
[3] https://projecteuclid.org/euclid.aos/1176344136
[4] https://arxiv.org/abs/1706.05148

--------------------------------

Notes and typos:

- In the introduction, "Deep clustering network network (DCN)", the word "network" is repeated
- After equation 5, "... where \pi( . ) denotes the parent node ...", the "pi" symbol does not appear in the equation at all, neither in the following equation, so I guess this part of the sentence should be removed
- In section 3.3, you write that you define 5 operators, but follow by listing 7 (NI, ND, SI, SD, NR, PO and UP)
- In section 4.1, I believe W lives in R^(10x4) not R^(10x2)
- In section 4.5 the acronym "MoG" ("Mixture of Gaussian" I guess) is used without being introduced previously

---

> ### Author Response · Authors · 2018-11-23
> **Authors' response to reviewer 3**
>
> Q: What gains does the hierarchical structure on the Y variables provide?
> A: First, we show that the models actually learned conditional dependencies on the Y_i variables. For the MNIST dataset, the conditional probability between identity facet Y_1 (x-axis) and pose facet Y_2 (y-axis) is shown in Fig. 8 in the Appendix of the latest version of the paper just uploaded. It can be seen that a cluster in Y_1 facet could correspond to multiple clusters in Y_2 facet due to the conditional probability, e.g. cluster 0, 4, 5, 11 and 12. However, not all clusters in Y_2 facet are possible for a given cluster in Y_1 facet.
> Second, we show that models with no links between Y_1 & Y_2 are inferior to models with links. We take the model structure produced by LTVAE, remove the links between Y1 & Y_2 and it becomes two independent Gaussian mixtures. We re-estimate the parameters of the resulting model, and evaluate the test-set log-likelihood of it. The test-set log-likelihood of such model in MNIST is -120, which is significantly lower than that of LTVAE (-83.67). It shows that the independent Gaussian mixtures model does not fits the data as good as LTVAE, suggesting the importance of hierarchical structure on the Y variables. In general, latent variables are often correlated. For example, the social skill and academic skill of a student are often correlated. Therefore, it’s better to model this relationship to better fit the data
>
> Q: How do different facets interact with each other? How are the samples from different clusters of facet 2 when facet 1 is fixed to a particular cluster? Assuming the learned dependency is that Y_1 is the parent of Y_2, does the interpretation of each value of Y_2 change depending on the value of Y_1?
> A: As mentioned earlier, different facets interact with each other through conditional probabilities. Different facets control different attributes of the data. For example, in Fig. 5(c), we fix the cluster in facet 1 (identity), and show images from different clusters from facet 2 (pose). It can be seen that different poses of the same digits are clearly recognizable. To further validate it, we show some conditional image generation results in Fig. 9 in Appendix. For an input image, we fix the z variables in identity facet and sample the z variables from different clusters in the pose facet in order to generate new images. As it can be seen, the poses of the input digits are changed in the samples generated, while the identity remains the same.
>
> It should be noted that, as is well-known, the edge orientations in LTMs are unidentifiable, meaning that the model with Y_1 being the parent of Y_2 is equivalent with the model with Y_2 being the parent of Y_1. The dependencies determine how probable for each value of Y_2 when Y_1 takes a certain value. The interpretation of each value of Y_2 is consistent. For example, in Fig. 5(c) and Fig. 9, the same cluster in facet 2 for different digits show the same pose.

---

> > ### Comment · AnonReviewer3 · 2018-11-24
> > **Most concerns adressed**
> >
> > Thank you for your explanations and the associated addition to your paper. This resolves most of the concerns I had noted in my initial review. The fact that LTVAE seems to learn a latent structure in which the Y_i are highly dependent on each other is quite an interesting point, and it indeed matches the intuition we can have about the data in general. I'm raising my rating following these improvements.
> >
> > I am however not really convinced by your approach to empirically compare the performance of LTVAE wrt an equivalent model where the Y_i are independent. The immediate reason is that your reported test log-likelihood (-120) is much worse than that of a regular VAE as per your paper (-84.9), which is quite surprising as the regular VAE would be a special case of both LTVAE and this independent approach. This leads me to believe that your approach of first training using LTVAE and then amputating it has an issue. This likely makes the model fall into some local minimum that it would not have reached if trained from scratch, especially given TLVAE seems to learn a structure where the latent variables are highly dependent.

---

### Official Review · AnonReviewer2 · 2018-11-04
**Well written, carefully thought through, and very interesting paper with impressive empirical results**

**Rating:** 8
**Confidence:** 4

**Review:**

This paper introduces a new VAE model, the latent tree VAE (LTVAE), which aims to learn models with multifaceted clustering, that is separate clusterings are enforced on different subsets of the latent features.  This is achieved using a tree-structured prior on a set of discrete "super latent variables" (Y_1,...,Y_L) that identify which cluster the datapoint falls into for each separate facet (i.e. there is a separate clustering associated with each Y_n).  The subset of the standard latent variables z then form a Gaussian mixture model (GMM) for each Y_n.   Both the structure of this setup (i.e. the associated graphical model) and the parameters (i.e. means and variances of the clusters) are learned during training.  This introduces a number of computational challenges not usually seen in for VAE training, for which, seemingly well thought through, novel schemes are introduced, most notably a message passing scheme for calculating gradients of the log marginal p(z).

Overall I think this is a very good paper.  The exposition of the work is, for the most part, very good - the paper was a pleasure to read.  I think that the key idea is novel and adds something unique and useful to the literature, I thus think it is work which will be of substantial interest to the ICLR community.  The quality of the paper is also very good: algorithmic details seem to have been well thought through and the experimental evaluation is above average, both in terms of apparent performance and in the breadth of experiments considered.  I would very much like to see this work accepted to ICLR and I think that the extra use of space over 8 pages in the submission is justified.  However, I do have some questions and concerns that I would like to see addressed in the rebuttal period and I may lower my score if they are not.

The key issues I would like to see addressed further discussion on are:
a) There is no discussion about what is done for the encoder in the paper.  This is surely a very important consideration here as if the encoder is not expressive enough, this will impact the learned models.  For example, the dependency structures of the latent space induce particular dependencies in the posterior that must be carefully handled to avoid harming the learning (see e.g. https://arxiv.org/abs/1712.00287).
b) I would like to see some numerical results for the similarity between the different clusterings that are learned.  A lot of the novelty of the work rests on being able to pick up different clusterings with the different facets.  However, the results suggest that the clusterings may actually have very significant overlap and so this should be quantified.
c) The approach is presuming substantially slower than a setup where the structure is pre-fixed.  I think it is fine even if there is a big slow down, but I would like to see timing information so that the reader can assess how much higher the time cost is.
d) As far as I can tell (sorry if I have made a mistake), the presented results are from single runs.  I would like to see information about the variability across different runs so that the fragility of the approach can be assessed.
e) I would like to see more justification for having a dependency structure between the Y's, ideally both in motivating this choice and in experimental evaluation to check it (more generally ablation studies for different components of the algorithm would improve the paper).  Might it be possible to use this in a way the encourages the different clusterings to be distinct from one another?

Other comments:
1) Though the writing is generally very good, there are a few exceptions:
- The second paragraph in the intro becomes a list of related work from the point where DEC is introduced.  This should be moved to the related work to improve the flow (just cite those papers at the end of the first sentence in the third paragraph) and it would be good for it to be less of a list of separate things and more something that puts the current work in the context of other approaches.
- The paragraph after Eq 3 needs some rewriting
- The explanations around and including equations 5 and 6 were quite poor: \pi is referred to but not used, it is not made clear that that g is the gradient of log p(z) instead of p(z), use brackets for the log in Eq 6 to avoid ambiguity
2) The reference formatting is wrong (i.e. cite is used everywhere instead of citep)
3) I thought the motivation for the approach in the intro was very good
4) As the seemingly most related work, it would be good to elaborate more on the Goyal et al paper and the differences of your approach to theirs.  Is there a reason this is not used as a baseline in the experiments?
5) I could not understand the step from the gradient to the gradient of the log in Eq 6.  Is this because p(y_b|z) = f(y_b) Norm(..)?
6) The text in figures 2 and 3 is too small and difficult to make out.
7) I think it is misleading to talk about p(z) as being a marginal likelihood and would use the term marginal prior, or just marginal, instead.
8) I thought Figure 4b provided a nice demonstration.
9) Is there a reason that log likelihood / ELBO scores are only provided on MNIST and only for the LTVAE / VAE?  I might be wrong, but I thought at least some of the other baselines provide this and those results presumably already exist as a side effect from calculating the clustering scores?  Relatedly, I'm aware that a previous version of this work included estimates of the normalized mutual information -- is there any reason these are no longer included?
10) Did the larger dimensional latent spaced used for the qualitative results improve or worsen the performance of previous metrics?

Minor points / typos
- mehod -> method
- of generation network -> of the generation network
- brackets in eq 7
- MoG not defined in section 4.5

---

> ### Author Response · Authors · 2018-11-23
> **Authors' response to reviewer 2**
>
> Key issues:
> Q: Discussion about the encoder to handle the dependency structures of the latent space.
> A: For the inference network (encoder), we simply take the mirrored neural network structure of the generative network. The neural network structure is fully-connected or convolutional. In this way, the inference network is not minimal as in (Webb et al., 2018), but matches the capability of the generative network. Since the prior in LTVAE is more complex, presumably the learning of the models could benefit from important weighted sampling as proposed in IWAE (Burda et al., 2016).
>
> Q: Similarity between different clustering that are learned.
> A: The similarity between different clusterings could be quantitatively described by normalized mutual information (NMI). We compute the NMI between two kinds of partitions. For MNIST dataset, the NMI between identity facet and pose facet is 0.682. Note that the NMI between identity facet and ground truth is 0.825. This suggests that the two facets (identity and pose) have sufficient non-overlap and that they capture different aspects of the data. We show more detailed NMI results between the found facets for different datasets in Table 4 and Table 5 in the Appendix D of the updated pdf.
>
> Q: Computational time compared with prefixed structure.
> A: We show the computational time comparison in Table 3 in the Appendix C of the updated pdf. For LTVAE with fixed structure, we fixed the structure of the latent tree model to be a single Y connecting to all z’s, in which each z node consists of single z variable. As it can be seen that, the computational time of LTVAE with structure learning is indeed several times slower than that with fixed structure. For example, for MNIST dataset, it is two times slower than that with fixed structure. Most of the extra computational time is spent on the structure searching and parameter estimation of the candidate structures.
>
> Q: Variability across different runs.
> A: We show the variability of testset loglikelihood across different runs in Table 6 in Appendix E. As for the clustering accuracy, the deviations are around +/- 1%.
>
> Q: Justification for having a dependency structure between the Y’s.
> A: One facet of data corresponds to one latent variable. In general, latent variables are often correlated. For example, the social skills and academic skills of a student are often correlated. Therefore, it’s better to model this relationship to better fit the data.
> We further show that models with no links between Y_1 & Y_2 are inferior to models with links. We take the model structure produced by LTVAE, remove the links between Y1 & Y_2 and it becomes two independent Gaussian mixtures. We re-estimate the parameters of the resulting model, and evaluate the test-set log-likelihood of it. The test-set log-likelihood of such model in MNIST is -120, which is significantly lower than that of LTVAE (-83.67). It shows that the independent Gaussian mixtures model does not fits the data as good as LTVAE, suggesting the importance of hierarchical structure on the Y variables.
>
> Other comments:
> Q: Differences of the proposed approach to that in Goyal et al.
> A: The difference lies in that there is no multi-facet concept in Goyal et al. An image is assigned to one and only one path of the nCRP tree. It is still one partition over the data, only that the partitions in upper level are more general partition, while those in lower level more fine-grained. While the proposed approach of our paper, for example, makes one partition based on identity, while makes another partition based on pose. Furthermore, a leaf node in Goyal et al. is always the whole z, and it is just that for different data the z belongs to different leaf node. While for our approach, a parent node may only connect to a subset of z. Besides that, Goyal et al. mainly deals with video data, and we are unable to obtain its implementation for comparison.
>
> Q: Derivation of Eq.6.
> A: Eq. 6 is to compute the gradient of log p(z) w.r.t z_b. The gradient to the gradient of the log step is \sum_{y_b} 1/p(z) * \partial f(y_b) N(z_b | y_b) / \partial z_b  =>  \sum_{y_b} 1/p(z) * f(y_b) N(z_b | y_b) * \partial \log f(y_b) N(z_b | y_b) / \partial z_b . Note that 1/p(z) * f(y_b) N(z_b | y_b) = p(y_b|z) due to Eq. 5 and that f(y_b) is the collection of all terms not containing z_b.
>
> Q: Loglikelihood only provided on MNIST and only for LTVAE/VAE.
> A: We provide testset loglikelihood for other datasets in Table 6 in Appendix E. Note that the rest of the baselines are not probabilistic model and cannot compute loglikelihood. In addition to LTVAE/VAE, we add the loglikelihood results of IWAE (Burda et al., 2016) for comparison.
>
> Reference:
> Webb, Stefan, et al. "Faithful Model Inversion Substantially Improves Auto-encoding Variational Inference." NIPS (2018).
> Burda, Yuri, Roger Grosse, and Ruslan Salakhutdinov. "Importance weighted autoencoders." ICLR (2016).

---

> > ### Comment · AnonReviewer2 · 2018-11-23
> > **Quite a few concerns not updated in the paper itself**
> >
> > Thank you for your above clarifications and for the new results added to the appendices.  However, I am little surprised that many of the concerns raised were only considered in your above response, but not the paper itself, particularly issues a) and e).  I really think the paper itself needs updates to address these, while the extra distinct to the Goyal et al paper above should be added and extra clarification around eq 6 is still very much needed in the paper itself.  In general, the updates seem very rushed.
> >
> > I also have concerns in your response to issue a): by "mirror" I presume that you mean you just flip the direction of all the dependencies?  If so your claim that this matches the capability of the generative network is untrue as demonstrated by Webb et al 2018.  This is a potentially serious issue as it could mean that the main driving force for the imposes structure is actually the encoder (rather than the generative model itself), which is currently completely glossed over in the paper.  Note that an unfaithful encoder prevents the variational bound becoming tight and so will indirectly impact the models learned.  I thus see it as extremely important that the paper at least provides some discussion on this issue.  More generally, the fact that the encoder is not discussed anywhere in the paper is a real issue.
> >
> > As things stand, I am planning to reduce my original score as I think the above is a very important issue (perhaps the most important one raised) and I do not think it has been at all addressed.
> >
> > Other comments
> > - I think it is important to add error bars to the clustering accuracy results, not just the log likelihoods.  These are actually the results that seem most likely to have high variance.  I appreciate this might be difficult to do before the end of the rebuttal period though.
> > - Figure 8 that has been added to the appendices needs cleaning up and a proper explanation providing.  More explanation for figure 9 would also help.
> > - Why is Table 6 added as a separate table rather than updating Table 1?  Why is there such a miss-match between the two VAE results?  Please also include error bars for IWAE.

---

> > > ### Author Response · Authors · 2018-11-25
> > > **Authors' response to reviewer 3**
> > >
> > > Thank you for the comments and suggestions. Now we have incorporated above responses in the paper itself.
> > > We agree with the reviewer that the inference network is indeed an important issue. The discussion on inference network is added in Paragraph 2 of Discussion section and we quote it here:
> > > In this paper, for the inference network, we simply use mean-field inference network with same structure as the generative network (Kingma & Welling, 2014). However, the limited expressiveness of the mean-field inference network could restrict the learning in the generative network and the quality of the learned model (Webb et al., 2018; Rainforth et al., 2018; Cremer et al., 2018). Using a faithful inference network structure as in (Webb et al., 2018) to incorporate the dependencies among latent variables in the posterior, for example one parameterized with masked autoencoder distribution estimator (MADE) model (Germain et al., 2015), could have a significant improvement in learning. We leave it for future investigation.
> > >
> > > The discussion on the justification of dependencies structure on Ys is added in Paragraph 1 of Discussion section.
> > > The extra distinct to the Goyal et al paper is added at the end of Related works section.
> > > Due to the space limitation, we added the detailed derivation in Appendix F, and reference it around Eq. 6. We hope it’s ok.
> > > The additional results of loglikelihood in Appendix have been merged to Table 1. We keep Table 6 in Appendix for now for the rebuttal of other reviewers.
> > > Error bars will be added to Table 2 once available.
> > > Appendix has been polished with explanations.

---

> > > > ### Comment · AnonReviewer2 · 2018-11-26
> > > > **Concerns now addressed**
> > > >
> > > > Thank you for the speedy further revision - I appreciate that I have been a little demanding! Though I think the paper could still be refined a bit further on the points I and the other reviewers have raised for its final version (in particular I think there are still some open questions relating to the encoder), I am now happy that my concerns have been adequately considered and will stick with my original high score. I am glad that there seems to be agreement between the reviewers for accepting this paper as I think it will be of great benefit to the ICLR community. Great work!

---

### Meta-Review · Area_Chair1 · 2018-12-18
**Interesting approach to completely learn a structured prior, but reaching worse likelihood**

**Confidence:** 3
**Recommendation:** Accept (Poster)

**Metareview:**

A well-written paper that proposes an original approach for leaning a structured prior for VAEs, as a latent tree model whose structure and parameters are simultaneously learned. It describes a well-principled approach to learning a multifaceted clustering, and is shown empirically to be competitive with other unsupervised clustering models.
Reviewers noted that the approach reached a worse log-likelihood than regular VAE (which it should be able to find as a special case), hinting towards potential optimization difficulties (local minimum?). This would benefit form a more in-depth analysis.
But reviewers appreciated the gain in interpretability and insights from the model, and unanimously agreed that the paper was an interesting novel contribution worth publishing.